# Atrial Fibrillation in Athletes—Features of Development, Current Approaches to the Treatment, and Prevention of Complications

**DOI:** 10.3390/ijerph16244890

**Published:** 2019-12-04

**Authors:** Evgeny Achkasov, Sergey Bondarev, Victor Smirnov, Zbigniew Waśkiewicz, Thomas Rosemann, Pantelis Theodoros Nikolaidis, Beat Knechtle

**Affiliations:** 1Department of Sports Medicine and Medical Rehabilitation, Sechenov First Moscow State Medical University (Sechenov University), Moscow 119991, Russia; 2215.g23@rambler.ru (E.A.); sabondarev@yandex.ru (S.B.); z.waskiewicz@awf.katowice.pl (Z.W.); 2Department of Hospital Therapy, Saint-Petersburg State Pediatric Medical University Ministry of Health of Russia, Saint Petersburg 194100, Russia; vs@tdom.biz; 3Institute of Sport Science, Jerzy Kukuczka Academy of Physical Education, 40-065 Katowice, Poland; 4Institute of Primary Care, University of Zurich, 8091 Zurich, Switzerland; thomas.rosemann@usz.ch; 5Exercise Physiology Laboratory, 18450 Nikaia, Greece; pademil@hotmail.com; 6Medbase St. Gallen Am Vadianplatz, 9001 St. Gallen, Switzerland

**Keywords:** Cardiomyopathy, atrial fibrillation, myocardial stiffness, echocardiography, SPECT, PET, athletes, oral anticoagulants, over-training syndrome, pathological athlete heart

## Abstract

Atrial fibrillation (AF) is one of the most common types of cardiac arrhythmias. This review article highlights the problem of the development of atrial fibrillation in individuals engaged in physical activity and sports. Predisposing factors, causes, and development mechanisms of atrial fibrillation in athletes from the perspective of the authors are described. Methods of treatment, as well as prevention of thromboembolic complications, are discussed. Directions for further studies of this problem and prevention of complications are proposed.

## 1. Introduction

Atrial fibrillation (AF) is one of the most common types of cardiac arrhythmias (CA). More than a third of hospital admissions to cardiology hospitals are associated with the development of AF and its complications. The detection rate of this CA is about 0.4% in the general population, <1% in patients <60 years old and >6% in patients >80 years old [1]. The mortality among patients with AF is almost two times higher than in patients with sinus rhythm [2]. The incidence of ischemic stroke in patients with AF of a non-rheumatic etiology averages 5% per year, which is 2–7 times higher than that of individuals without AF.

## 2. Prevalence

According to different authors, the prevalence of AF in athletes is 2–10 times greater (five times on average) than in the general population. The magnitude of the prevalence of AF is truly large. According to published data, AF mostly occurs in individuals who train the quality of endurance. According to British researchers in England, more than two million people a year take part in marathons. According to the results of an electrocardiogram (ECG) control, 5% to 10% of the athletes among middle-aged marathon runners (over 35 years old) suffer from AF [3]. Most authors examined only men. However, in studies involving athletes of both sexes, no gender differences were detected [4,5,6,7].

AF reduces the quality of life and the ability to exercise in athletes [7]. Meanwhile, this is a phenomenon of extremely high overvoltage of the body and is a serious disease [8,9,10,11]. Given the frequency of formation of thromboembolic complications, the problem of AF in athletes is particularly relevant. Paget–Schroetter syndrome, injuries of limbs, dehydration, and compression commonly occur in sportsmen. Under such conditions, the incidence of AF is more than four people per 100,000 cases. In this group, diagnosed pulmonary thromboembolia occurs at a frequency of two per 100,000 athletes, and the mortality rate reaches 6–7% [12,13].

## 3. Etiology

Most researchers have noted a direct connection between the development of paroxysmal AF and the duration and intensity of physical exertion. Physical activity for more than 10 h per week for at least 10 years is significant in terms of the risk of arrhythmia.

From the standpoint of various authors, a large number of non-genetic causes (or reasons that have no proven connection with genetic abnormalities) and their combinations with known genetically determined conditions play a role in the development of atrial fibrillation in athletes.

Genetic determinants and acquired diseases are among the factors predisposing to the development of AF. According to European experts in the field of sports and preventive cardiology, prolonged physical overload can contribute to the manifestation of arrhythmogenic and dilated cardiomyopathy, and is also a risk factor for sudden cardiovascular death [9,14]. Genetically determined factors include genes encoding the development of hypertrophic cardiomyopathy (HCM) gene 5′-AMP-activated protein kinase subunit gamma-2 5′-AMP-activated protein kinase subunit gamma-2 5′-AMP-activated protein kinase subunit gamma-2 (PRKAG), tachy-brady syndrome, gene receptor adrenérgico beta 1 (ADRB1), production of atrial natriuretic factor, activity of the sodium and potassium channels of cardiomyocyte membranes, and gene encodes the synthesis of the α-subunit of the voltage-gated sodium channel of type V (SCN5A) [15,16,17]. From the position of American researchers at the Mayo Clinic, the role of genetic factors is leading in the formation of almost all arrhythmias in athletes [18]. This statement certainly deserves attention; however, on the one hand, this requires further confirmation, and on the other hand, a number of changes occur in the controlling genes, whose activity can be modulated by environmental factors and, consequently, sports stress [19].

Pathological changes in the heart develop against the background of irrational training, which, according to the official position of the European College of Sports Sciences, is the result of nonfunctional overreaching. This can be the result of minor disorders and last for weeks or longer and lead to deeper disorders in the form of over-training syndrome, which can last for months and be in itself the cause of a violation of a sports career.

With these disorders, there is a violation of the sympathetic and parasympathetic regulation of the body, manifested by fatigue, weakness, decreased results, bradycardia (in order to reduce the activity of sympathetic regulation), tachycardia, insomnia, irritability (typical for reducing the activity of parasympathetic regulation). Such abnormalities occur in 30–55% of athletes with a combination of the absence of reasonable periods of rest during training, as well as in combination with other known diseases, such as a cold [19].

The development of this syndrome is based on several processes. Decreased glycogen storage during heavy exercise leads to a lack of energy synthesis, disruption of central nervous regulation due to depletion of tryptophan and increased sensitivity to serotonin, oxidative stress due to depletion of antioxidant systems and the formation of a large number of free radicals, a decrease in the level of glutamine in muscle cells, and disorders of hypothalamo-adrenal, adrenal and gonadal regulation. The level of pro-inflammatory cytokines also rises in response to over-training with damage to muscle mass. Interleukin 1 beta (IL-1b), interleukin 6 (IL-6) and tumor necrosis factor alpha (TNF-α) contribute to the inhibition of testosterone synthesis and lower levels of glycogen and glutamine [20,21]. By reducing the level of glycogen, glutamate contributes to a violation of the energy state of the body and the myocardium in particular; this can lead to a violation of protein synthesis and a violation of the state of the outer cell membrane and electrolyte cell imbalance. Pro-inflammatory cytokines, free radicals, and adrenaline contribute to the activation of fibrosis of muscle tissue and the myocardium. All of the above in addition to the development of over-training syndrome is the basis for the formation of AF.

The pattern of development of AF is determined by the degree of dilatation, an increase in the volume of the left atrium (from baseline values) and the number of training hours for the entire sports career. For those who practice less than 1500 h, the average dilatation of the left atrium was 24%; for those who practice from 1500 to 4500 h, the average dilatation was 40%; and for those who practice more than 4500 h, the average dilatation was 83% [12,13,22,23]. It should be noted that these indicators are very conditional. This is due to the fact that modern sport requires simultaneous training of different qualities, as well as each athlete individually dosing the intensity of the loads. Obviously, one hundred hours more work in anaerobic mode leads to severe energy, electrolyte, thermal and other disturbances. Such a disorder can result in cardiomyocyte damage and AF.

Athletes are characterized by an increased activity of the sympathetic nervous system during the daytime in combination with pronounced vagotonia at rest, volume loading, myocardial hypertrophy and dilatation of chambers, which provides as the basis for LDCs [14,24,25]. Mont and co-authors [22] performed a prospective analysis of 32 studies conducted in various centers with more than 2000 athletes of various motor orientations in comparison with the same number of people leading a sedentary lifestyle. All studies reported a persistent or paroxysmal form of AF in all participants [13]. At the same time, there were no significant differences in the degree of hypertrophy and dilatation of the left ventricle and dilatation of the left atrium in the studied groups. This fact was quite remarkable: individuals with a sedentary lifestyle developed atrial changes under the influence of diseases, and athletes under the influence of significant overloads. The authors noted a characteristic tendency for a paroxysmal form of AF in athletes. These facts require more extensive research and observation. It is likely that people with a permanent form of AF could not be included in the study, because they were unable to perform long-term workloads and were not included in the relevant teams.

However, not only acquired diseases and genetic factors influence the development of AF in athletes. The development of AF is affected by the duration of a sports career, the intensity of classes, and the athlete’s age. It was shown that among aged cyclists who continued training (66 ± 7 years), AF occurs eight times more often than in the control group. This indicates the influence of the nature of training (stamina training is more significant) and the age of athletes on the frequency of AF development [25]. However, it is obvious that sports and professionalism, including participation in competitions, can have a specific direct impact on the development of AF. This fact requires further studies.

The intensity of physical training, along with its duration, plays a role in the development of the predisposition to AF. This conclusion has allowed us to carry out a number of studies on the frequency of detecting AF in multidirectional sports. Ex-professional cyclists and golfers who continued to train in endurance and complex coordination at an average age of 66 years showed significant differences in the incidence of AF. In cyclists, AF was detected in 10% of cases, whereas golfers did not have any paroxysms [25].

Along with the described hereditary and acquired factors for the development of AF in a prospective study, based on a survey of 1160 44-year-old male athletes, Mont et al. [22] detected a higher incidence of AF in such a group of individuals. The prevalence of AF among active athletes was five times greater than among people with a sedentary lifestyle. The authors explained this fact with a higher myocardial mass, the size of the left atrium, the thickness of the left ventricular myocardium, and the vagal activity at rest. A similar conclusion regarding the frequency of AF development, as well as the mechanisms of its development, was made by a further prospective representative Kasper study based on a survey of 52,755 athletes with detected AF in 959 participants [25].

Thus, it is obvious that AF is determined by a number of known factors acquired during life, which can more clearly manifest in the presence of a hereditary predisposition, as well as under the influence of other factors, in particular related to sports stress. The likelihood of developing AF is higher at an older age, in individuals training endurance, and in individuals undergoing irrational training. It does not directly depend on professionalism, but depends more on irrational training. Acutely developed disturbances in the activity of the heart is the key mechanism of AF development in people with rare sports loads. Such disorders may include oxidative stress, impaired macroerg synthesis, impaired formation, delivery of glucose and fatty acids, and tissue hypoxia.

The development of obesity, oxidative stress, an increase in the level of rho-kinase and cardiotropin 1 can be attributed to non-genetic factors in the development of AF in athletes [26,27,28,29]. Also, in response to stress, a number of processes occur under the influence of catecholamines, such as stimulation of lipid peroxidation, lipase and phospholipase, and increase in cAMP synthesis in cardiomyocyte membranes, activation of protein kinase A, accumulation of Ca2+ in accumulated calcium ions, calmodulin, cell stimulation of glycolysis and inhibition of glycogen re-synthesis [17]. Recent studies led by Professor D. Corrado [27] indicate the role of the pathology of ryanolodine receptors on cardiomyocyte membranes, which contribute to the accumulation of calcium ions in cardiomyocytes and also play a role in the development of cardiac rhythm disturbances [30].

The factors contributing to the formation of AF include obesity and the development of arterial hypertension. According to recent research [31], the prevalence of arterial hypertension among young athletes up to 35 years old is approximately 5.6%. Among adolescent children involved in sports, essential arterial hypertension of grade 1–3 is two times more common than in the same group of non-athletes. According to researchers, the development of arterial hypertension leads to pathological atrial remodeling in the form of dilatation and fibrous degeneration. Such changes occur under the influence of volume overload, which increases during exercise, with the background of tachycardia, and due to ventricular diastolic dysfunction. Ventricular diastolic dysfunction is a result of myocardial remodeling during arterial hypertension and takes part in the formation of the athlete’s pathological heart [32,33,34,35].

The problems of super-nutrition and obesity in athletes is as relevant as for people with a sedentary lifestyle. Athletes who have finished their career are three times more likely to be obese. According to large-scale preventive examinations in the United States, the prevalence of this pathology among young athletes is approximately 23.5% [33,34]. An increased expression of pro-fibrotic cytokines (such as transforming growth factor (TGF)-b), the development of structural remodeling and interstitial fibrosis of the myocardium of the atria, changes in the electrophysiological properties of cardiomyocytes, chronic systemic inflammatory status, oxidative stress, and increased activity of the renin–angiotensin–aldosterone system leads to the progression of AF in overweight and obese individuals [35,36]. Along with the problem of being overweight, an important risk factor of AF is a carbohydrate dysmetabolism, which is often found in the described group of patients and which contributes to the development of so-called diabetic cardiomyopathy. Carbohydrate dysmetabolism contributes to the regular use of high-energy nutrition and stimulants, including anabolics [8,29,37,38,39,40,41].

The above information from the literature is very important. However, in practice, there is a serious problem with the differential diagnosis between over-training, obesity, diabetes and myocarditis. Myocarditis is one of the main reasons for AF paroxysms, and its diagnosis is extremely difficult. According to the recommendations of the European Society of Preventive Cardiology, in the presence of AF paroxysms in young people, including athletes, we must conduct an examination to exclude inflammatory myocardial damage. There are many unexplored points in this matter (i.e., when antibodies and signs of cardiomyocyte damage are formed both with myocarditis and with myocardial overstrain). In our opinion, the anamnesis, magnetic resonance imaging (MRI) data with signs of myocardial edema, and the results of a myocardial biopsy with a proven inflammatory process will help to understand this situation [42,43].

## 4. Physiological Interpretation

Studying the problem of AF in professional sport is fraught with difficulties. This is mostly due to a combination of various factors, such as individual fitness, talent, and the dedication of athletes. In this regard, there are no clear parameters for heart cavity size and ventricular wall thickness nowadays. However, according to most authors, the left ventricular end-diastolic diameter in athletes in B -modal mode using short axis views is up to approximately 54 mm, while the thickness of the left ventricular myocardium is not more than approximately 13 mm. The diameter of the left atrium is up to approximately 37–50 mm or up to 36 mL/m^2^ in men and approximately 33 mL/m^2^ in women; the diameter of the right atrium is up to approximately 19–25 mL/m^2^ in men and approximately 15–25 mL/m^2^ in women. At the same time, indices of left ventricle diastolic function remain at the level of reference values: the deceleration time of the left ventricular outflow tract is 200–300 ms; isovolumic relaxation time is less than 100 ms; the early diastolic component of the medial mitral ring movement in tissue Doppler mode, peak e >10 cm/sec; the ratio of the peaks of the early transmitral blood flow, measured by a constant wave Doppler (peak E) and tissue Doppler by analyzing the amplitude of motion of the medical part of the mitral ring (peak e), E/e >6; and the systolic component of the mitral ring movement in tissue Doppler mode, peak S > 9 [26,27,32,33,44].

Dilatation of cardiac cavities above the specified values, and importantly, appearance of signs of impaired myocardial relaxation recorded by the method of constant-wave Doppler ultra-sonography (which is one of the components of atrial remodeling) develop in over-trained athletes [33,41]. Thus, the severity of diastolic dysfunction as well as the volume and intensity of physical activity plays a role in the formation of increased atrial size, which is a predisposing factor for AF formation and fixation.

AF can be induced by sympathetic or parasympathetic innervation. Induction by the parasympathetic nervous system is predominant in athletes. Such a pattern is observed in more than 68% of AF in athletes, regardless of the athletic quality trained [33,45,46]. An increase in vagus nerve tonus may indicate a sympathetic nervous system breakdown against over-training [33]. An increased vagus nerve tonus initiates AF by creating a macro-reactive increase in the dispersion of the atrial refractory period [33,34]. In addition, reflex influences play a certain role in the provocation of AF. One of the predisposing factors is the presence of gastritis and reflux esophagitis [35].

According to Benito et al. [26], the formation of atrial and ventricular myocardium degeneration is typical for athletes of all age groups against the background of increased production of fibrosis factors such as fibronectin 1, procollagen 1 and 3, transforming growth factor B1, matrix metalloproteinase 2, and tissue inhibitor of metalloproteinase 1. According to the authors, these mechanisms aggravate and accelerate in athletes. The level of profibrotic factors in athletes rises with increasing athletic experience and age [26]. Similar data were obtained by Wolk and co-authors [13] based on a survey of 45 elite athletes with AF. According to the results of echocardiography and MRI, a significant myocardial fibrosis was detected in these patients compared with an identical group of people not involved in sports. It was demonstrated that collagen metabolism markers (such as metalloelastase) were higher in master athletes compared to sedentary individuals. MRI with gadolinium, performed for the detection of fibrous degeneration of the ventricular and atrial myocardium, showed similar data [41,45].

## 5. Diagnosis

For athletes, paroxysmal atrial fibrillation for five minutes or longer is clinically significant [31,34]. Naturally, such duration of paroxysms requires appropriate approaches to the diagnostic process. According to the official position, the main method of AF diagnosis is electrocardiography (ECG) in 12 leads and 24-h ECG monitoring. In athletes, paroxysmal AF often occurs either in the training process or after training or competition. Specialists can perform and correctly interpret the ECG in such conditions only with the use of telemedicine technologies. This is reflected in the relevant American and European recommendations [30,41]. According to the experts, all patients should have an ECG as part of the survey; in 19% of cases, ergometry was used; in 5%, 24-h long-term ECG was used; in 5%, echocardiography was used; and in 1%, cardiac catheterization and magnetic resonance diagnostics were used [42,47]. The above-mentioned volume of examination of athletes has reduced the mortality rate from 4% per 100,000 people to 0.4% over the past few years. Predisposing factors for AF in athletes are summarized in Table 1.

## 6. Treatment

The nature of the development of arrhythmia can be an argument for the preferred choice of treatment modality. The treatment of AF in athletes firstly begins with a complete cessation of physical stress, because paroxysmal AF in athletes is mostly not protracted and such tactics sometimes prove to be sufficient [41,45]. In the absence of the effect of the termination of the load, according to the currently accepted approaches, electrical or medical defibrillation is performed [44]. Drug therapy with Flecainide or Propafenone is usually prescribed [44,47]; beta-blockers such as Sotalol are the background therapy. Vagal AF is effectively treated by pulmonary vein isolation. According to the recommendations of the European Society of Cardiology, together with the European Arrhythmology Society, the following approaches are applicable for athletes: the principle of a pill in a pocket (class recommendation 2a, level C) and catheter ablation. The latter technique is preferred.

The method of transcatheter isolation of the pulmonary veins during atrial fibrillation allows for the saving of the sinus rhythm and avoidance of possible complications without the use of potentially dangerous anti-arrhythmic drugs. According to the majority of experts, an athlete after successful catheter ablation can be allowed to compete in any sport after 4–6 weeks in the absence of relapse and the absence of atrial fibrillation when performing an electrophysiological study [4,42,43,44,45]. The use of beta-blockers can be recommended for athletes both for the prevention of the development of relapse of atrial fibrillation and for the prevention of tachysystole. Also, the use of beta-blockers is etiopathogenetic with respect to reducing the damaging effect of adrenaline and norepinephrine on the myocardium. When prescribing beta-blockers with non-selective and selective action, it is necessary to take into account the possible contraindication to the use of these medicines in a particular sport. Surgical treatment of atrial fibrillation is preferred due to its high efficiency, as well as contraindication to the use of anti-arrhythmic drugs on loads and contraindication of beta-blockers in a number of sports. Beta-blockers are drugs that slow down atrioventricular conduction, the frequency of sinus rhythm and reduce the contractility of the heart. Therefore, the prescription of these drugs should take into account the possible negative effects and the change in standard approaches to control the reaction to training in terms of heart rate.

AF treatment should take into account the development of over-training syndrome. Therefore, athletes with AF, depending on the stage of development of the pathology, should have the necessary rest and receive pharmaceutical and non-pharmaceutical treatment to improve sleep and the emotional sphere. Potassium, magnesium, calcium, and drugs that promote the synthesis of ATP in the mitochondria (exogenous phosphocreatine, trimetasidinum), such as trimethacidin, as well as drugs with an antioxidant effect, including vitamins E, C, B, can be recommended as concomitant therapy to normalize the energy and electrolyte balance of cardiomyocytes. A number of these drugs are prohibited for use by international anti-doping agencies; therefore, when prescribing such drugs, it is necessary to clearly maintain the primary documentation and draw up relevant documents on the therapeutic use of these medications. 

Performing any sports is contraindicated while taking ion-channel blockers such as Propafenone and Disopyramide because of the high risk of sudden death [47]. Permission to continue exercise is possible with the restoration of sinus rhythm and after two half-lives of the drugs taken have passed. It should be considered that in the case of the development of AF in individuals who train endurance, the effectiveness of isolation is lower at 34% (versus 48%) [46]. The authors also note a higher incidence of complications of both conservative and operative therapy in athletes compared with the rest of the population: 7.1% versus 4.3% [25]. Anticoagulant therapy in athletes should be prescribed, taking into account the risk of injury and bleeding. Treatment with new oral anticoagulants, as well as vitamin K blockers, has not yet been studied separately on the athletic population. According to the European recommendations for the general population, such drugs are prescribed with a focus on the Score for Stroke Risk Assessment in Atrial Fibrillation to CHA2DS2-VASc scale. It should be noted that in athletes, this method is carried out relatively rarely due to the lack of the number of corresponding points [47]. 

Nowadays, the new oral anticoagulants deserve special interest for use in athletes. The data on effectiveness, safety and clear advantages over the use of Warfarin are obtained. Dabigatran etexilate is a low molecular weight pro-drug that does not possess pharmacological activity. However, after ingestion, Dabigatran etexilate is rapidly absorbed by hydrolysis and catalyzed by esterases, and is converted to Dabigatran, a competitive reversible direct inhibitor of thrombin. Rivaroxaban is a direct specific selective inhibitor of the Xa coagulation factor, catalyzing the conversion of prothrombin to thrombin. It has no direct effect on thrombin, but regulates the formation of thrombin by inhibiting the Xa factor. It reaches the maximum plasma concentration within two to four hours after administration and its half-life is approximately 11–13 h. Apixaban is a direct specific selective Xa inhibitor that catalyzes the conversion of prothrombin to thrombin. The emergence of a specific antagonist to Dagibatran, Idarucizumab, currently expands the possibilities of using an oral anticoagulant in athletes and makes this method safer.

All these drugs have undergone clinical trials where their efficacy, safety, and advantages over Warfarin or a lower risk of stroke (for Dabigatran and Apixaban), or a lower risk of bleeding (for Rivaroxoban), have been proven. In this list, Dabigatran has a significant advantage, provided by its high tolerance. This substance has a specific antagonist, Idarucizumab, that neutralizes the effect of Dabigatran in life-threatening bleeding, for example, after a sports injury. This drug does not have any prothrombotic effects and allows patients to return to the previous therapy after 24 h [10,45]. Based on the data presented and in accordance with the recommendations of the European Cardiology Society and the Society of Preventive Cardiology, we developed a treatment plan for athletes with AF (Figure 1).

The current position on the treatment of AF is quite tough. However, another position was expressed at the 36th Conference of Bethesda Hospital [48]: Zipes and co-authors believed that athletes with self-stopping atrial fibrillation without symptoms could be admitted to all types of training and competition. This position requires further research, because according to the European Society of Preventive Cardiology, any AF with a risk of thromboembolia of 1 or more on the CHA2DS2-VASc scale requires the appointment of anticoagulants, especially for athletes with a high risk of blood clotting and activation of coagulation factors [48]. It is impossible to restore sinus rhythm in the presence of clinical manifestations of chronic heart failure; these athletes cannot be allowed to engage in sports that require the qualities of endurance, strength, or speed. According to the European recommendations for admission to competitive activity in 2010 of the Sport Cardiology Section of the European Association of Preventive Cardiology (2019), and taking into account the position of experts from the American Heart Association and American College of Cardiology, such athletes can be only admitted to low-intensity and non-dynamic sports. This significantly changes the lifestyle and motor activity of the athlete and requires not only cardiological, but also psychological support [42].

## 7. Conclusions

Hereditary and acquired factors play a role in the development of atrial fibrillation in athletes as in the main population. Among the latter, the presence of arterial hypertension, obesity, carbohydrate dysmetabolism, dilatation of atrial cavities, impaired ventricular diastolic function, and fibrous degeneration of the atrial myocardium are the most relevant. Along with this, the duration and the intensity of the sport and age (where older athletes are at higher risk) play important roles. For these individuals, the vagotonic form of this type of arrhythmia is most characteristic. To prevent the development of AF, a thorough history and complaints collection, vegetative balance assessment, ECG examination, 24-h ECG monitoring, echocardiography with assessment of ventricular diastolic function, stress tests, cardiac MRI with analysis of the degree of myocardial fibrous degeneration, as well as medical and genetic studies should be performed. In order to prevent embolic complications and sudden death in patients involved in sports, anticoagulant treatment should be carried out in the presence of risk factors. Given the low level of knowledge regarding the effectiveness of using new oral anticoagulants for the prevention of thromboembolism, it is advisable to conduct large-scale randomized clinical research to study their efficacy and safety.

The problem under consideration is relevant not only for athletes involved in classical sports, but also for professionals and amateurs who are gaining popularity in high-intensity or requiring long endurance sports, such as ultramarathons and triathlons. This is important, as older athletes are more likely to engage in such sports. Obviously, such athletes must undergo an in-depth examination. Trainers should clearly monitor the appearance of signs of fatigue and a breakdown of adaptation for the timely prevention of over-training syndrome.

## Figures and Tables

**Figure 1 ijerph-16-04890-f001:**
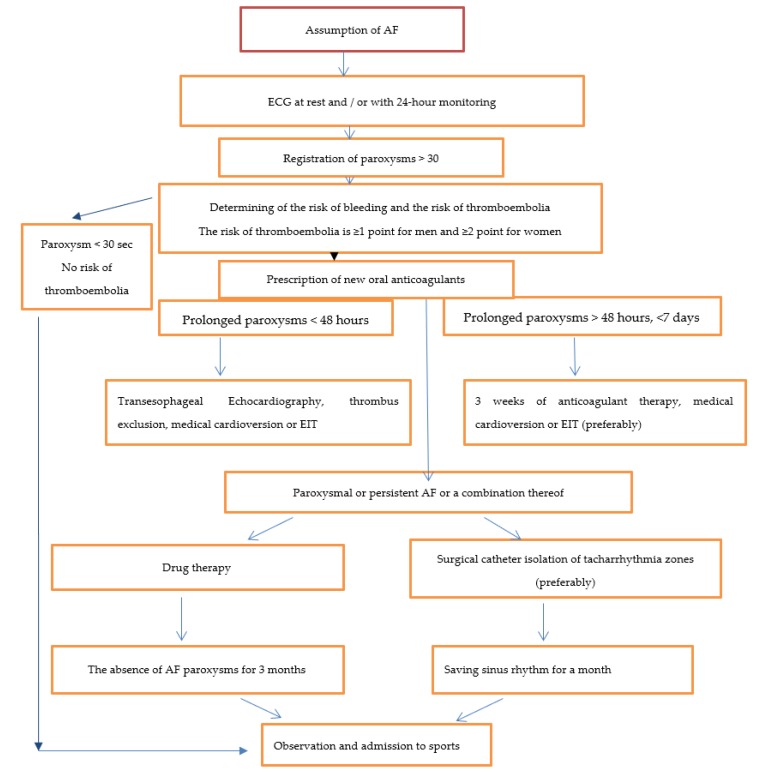
Treatment tactics for athletes with AF. (EIT—electropulse therapy).

**Table 1 ijerph-16-04890-t001:** Predisposing factors for atrial fibrillation (AF) in athletes.

Risk Factor	Index
Stamina training	Yes
Symptoms of non-functional overreaching	Yes
Symptoms of over-training syndrome	Yes
Overload of performance	Yes
Duration of physical activity	10 hours per week for 10 years or >4500 hours for a sports career
Family history	Indication of AF
Cardiomyopathy	Clinical and/or genetic confirmation
Obesity	Body mass index > 30
Impaired glucose tolerance	Yes
EchoCG size of LA	>36 mL/m^2^ for men 33 mL/m^2^ for women
EchoCG size of RA	>19–25 mL/m^2^ for men >13–25 mL/m^2^ for women
EchoCG size of LV wall	>13 mm
EchoCG time of isovolumetric relaxation of the left ventricle	<100 ms
E	>9 cm/sec
E/e	>6
The presence of arterial hypertension	≥1 degree
Myocardial mass	>400 mg
The systolic component of the movement of the mitral valve ring S	>9
MRI of the heart with gadolinium	Atrial fibroid degeneration
High degree of vagal activity according to 24-hour Holter monitoring and/or cardiac rhythmography at rest	Heart rate < 30 beats per minute AV block degree 2 type 2 or more Wave power increase F
Arrhythmias with daily monitoring of ECG	Atrial and/or ventricular premature beats > 2000 milliseconds; short paroxysms of atrial tachycardia
Increase in rho-kinase 2 and cardiotropin 1	Exceeding normal values
Increased production of fibrosis factors	Fibronectin 1, procollagen 1 and 3, transforming growth factor B1, matrix metalloproteinase 2, tissue inhibitor of metalloproteinase 1

CG = cardiogram; LA = left atrium; RA = right atrium; LV = left ventricle; MRI = magnetic resonance imaging; ECG = electrocardiogram; E = left ventricular early diastolic filling; e = early diastolic velocity of the mitral annulus; F = oscillation waves of RR intervals of ECG in the high-frequency region from 0.15 Hz to 0.4 Hz.; S—systolic component of the mitral ring movement in tissue Doppler mode.

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
