# Peer review of "Atrial Fibrillation in Athletes—Features of Development, Current Approaches to the Treatment, and Prevention of Complications"

_ijerph, 2019, doi:10.3390/ijerph16244890_

Round 1

Reviewer 1 Report

I have read with interest the paper entiltled: "Atrial fibrillation in athletes: Features of development, current approaches to the treatment and prevention of complications." This review deals with an issue that is quite common and of major importance among athletes of all sport disciplines and ages. 

Authors try to summarize available data, however there are some major limitations in this review:

1) There are several repetitions of data and expressions that puzzle the reader of the paper. Authors should definitely try to simplify the structure of their paper trying to avoid repeating data.

2) Better english language editing. Ankward expressions and syntax errors should be corrected.

3) The review paper should be accompanied by tables and figures. For example a Table summarizing all pathophysiology mechanisms predisposing to AF in athletes would be useful. Also a figure-algorithm guiding different steps of treatment wold also be a good idea.

4) Authors should critisize other papers and guidelines which they include in their references, rather that simply reporting their conclusions. A reader would be interested in their review if it "adds" sth in the literature and that is a critical approach to this issue.

5) References should be extended and filtered. For example references 16 and 17 are exactly the same paper published by different journals. Authors should choose one of them (the one published first). Additionally, there is no refernce to Bethesda Guidelines about arrhythmias in athletes.

Author Response

Comments and Suggestions for Authors

I have read with interest the paper entitled: "Atrial fibrillation in athletes: Features of development, current approaches to the treatment and prevention of complications." This review deals with an issue that is quite common and of major importance among athletes of all sport disciplines and ages. 

Authors try to summarize available data, however there are some major limitations in this review:

There are several repetitions of data and expressions that puzzle the reader of the paper. Authors should definitely try to simplify the structure of their paper trying to avoid repeating data.

Answer: We agree with the expert reviewer and eliminated thee repetition.

Better English language editing. Awkward expressions and syntax errors should be corrected.

Answer: We agree with the expert reviewer and we improved English and corrected errors

The review paper should be accompanied by tables and figures. For example, a Table summarizing all pathophysiology mechanisms predisposing to AF in athletes would be useful. Also, a figure-algorithm guiding different steps of treatment would also be a good idea.

Answer: We agree with the expert reviewer and made additions in the form of tables

Factors predisposing to the development of AF in athletes are summarized in table 1.

Table 1. Predisposing factors for AF in athletes.

Risk factor

Index

Stamina training

yes

Duration of physical activity

10 hours per week for 10 years or> 4,500 hours for a sports career

Family history

Indication of AF

Cardiomyopathy

clinical and / or genetic confirmation

Obesity

body mass index > 30

Impaired glucose tolerance

yes

EchoCG size LA

>36 мl/m2 for men

33 мl/m2 for women

EchoCG size RA

>19-25 мl/m2 for men

> 13-25 мl/m2 for women

EchoCG LV wall size

>13mm

EchoCG the time of isovolumetric relaxation of the left ventricle

<100 ms

Е

>9 cm/sec

E/е

> 6

The presence of arterial hypertension

≥1 degree

Myocardial mass

>400 mg

The systolic component of the movement of the mitral valve ring S

>9

MRI of the heart with gadolinium

Atrial fibroid degeneration

High degree of vagal activity according to 24-hour Holter monitoring and / or cardiac rhythmography at rest

Heart rate <30

AV block   degree 2 type 2 or more

Wave power increase F

Arrhythmias with daily monitoring of ECG

atrial and / or ventricular premature beats> 2000 short paroxysms of atrial tachycardia

Level rise of Ro-kinase 2 and Cardiotropin 1

exceeding Normal Values

Increased production of fibrosis factors

fibronectin 1, procollagen 1 and 3, transforming growth factor B1,matrix metalloproteinase 2,tissue inhibitor of metalloproteinase 1

E – left ventricular early diastolic filling

е – early diastolic velocity of the mitral annulus measured by tissue dopplerography in the medial part (EchoCG)

Holter – 24-hour Holter monitoring

F – oscillation waves of RR intervals in the high-frequency region from 0,15 to 0,4 Hz

Figure 1

Treatment tactics for athletes with AF                                                                                

AF suspicion

ECG at rest and / or with 24-hour monitoring

Registration of paroxysm> 30 ‘’

Determining the risk of bleeding and the risk of thromboembolism

The risk of thromboembolism is ≥1 point for men and ≥2 point for women

Observation and admission to sports

Пароксизм <30 sec

Риск тромбоэмболии 0

Prolonged paroxysm <48 hours

Prolonged paroxysm> 48 hours

Transesophageal Echocardiography, thrombus exclusion, medical cardioversion or EIT (предпочтительно) как можно раньше после

3 weeks of anticoagulant therapy, medical cardioversion or EIT (предпочтительно)

Private paroxysms or persistent AF

Drug therapy

Surgical catheter isolation of tacharrhythmia zones (preferred)

The absence of paroximes of AF for 3 months

Saving sinus rhythm for a month

Prescription of new oral anticoagulants

EIT- electropulse therapy

Authors should criticize other papers and guidelines which they include in their references, rather than simply reporting their conclusions. A reader would be interested in their review if it "adds" somethin in the literature and that is a critical approach to this issue.

Answer: We agree with the expert reviewer and made additions where we discuss controversial literature

It should be noted that these indicators are very conditional. This is due to the fact that modern sport requires training at the same time of different qualities, as well as each athlete individually doses the intensity of the loads. obviously, one hundred longer work in anaerobic mode leads to severe energy, electrolyte, thermal and other disturbances. Each such disorder can contribute to cardiomyocyte damage and AF.

The above literature data is very important. However, in practice, there is a very big problem of the differential diagnosis between the described diseases from overtraining to obesity and diabetes and myocarditis. Myocarditis is one of the main reasons for the development of AF paroxysms, and its diagnosis is extremely difficult. In the presence of AF paroxysms in young people, including athletes, we are required to conduct an examination to exclude inflammatory myocardial damage according to the recommendations of the European Society of Preventive Cardiology. In this situation, there remains a lot of unexplored moments when antibodies and signs of damage to the heart muscle are formed both with myocarditis and with myocardial overstrain. In this situation, from our position, an anamnesis, MRI data, where there will be signs of myocardial edema, and the results of a myocardial biopsy with a proven inflammatory process, can help [50].

All these drugs have undergone clinical trials where their efficacy, safety, and advantages over Warfarin or a lower risk of stroke, as for Dabigatran and Apixaban or a lower risk of bleeding, as for Rivaroxoban have been proven. In this list, Dabigatran has a significant advantage, provided that it is tolerable. Since it is to this substance that there is a specific antagonist, Idarucizumab, that neutralizes the effect of Dabigatran in life-threatening bleeding, for example, after a sports injury. This drug does not give a prothrombotic effect and allows you to return to the previous therapy after 24 hours [10,41, 46-49]. This fact significantly expands the possibilities of using new oral anticoagulants and athletes with a risk of injury [51]. This fact requires a subsequent review of the European recommendations for admission to the competition of athletes taking anticoagulants.

References should be extended and filtered. For example, references 16 and 17 are exactly the same paper published by different journals. Authors should choose one of them (the one published first). Additionally, there is no reference to Bethesda Guidelines about arrhythmias in athletes.

Answer: We agree with the expert reviewerю. We took into account the data of conferences and publications in Bethesda hospital.

The current position on the treatment of AF is quite tough. In this regard, there is a difference in the position expressed at the 36th Conference of Bethesda Hospital (Florida, USA, 2015), as set forth in an article by D. P. Zipes et al. The authors believe that athletes with self-stopping atrial fibrillation without symptoms can be admitted to all types of training and competition. This position requires further research, since from the position of the European Society of Preventive Cardiology, any AF with a risk of thromboembolism of 1 or more on the CHA2DS2-VASc scale requires the appointment of anticoagulants. Especially when it comes to athletes with a high risk of blood clotting and activation of coagulation factors [52].

Reviewer 2 Report

The submitted review of Achkasov et al. covers atrial fibrillation (AF) as a frequently type of cardiac complications in actual or former athletes. Unfortunately the authors did not differentiate different types of athletes and different kind of sports. But both are preconditions for the time and intensity of exercise / training load, which will lead to different kind of cardiac adaptations and (re-) modelling of the cardiac cavities, myocardial walls and tone.

Therefore the authors should also focus on:

Clear definitions of different types of athletes: Pro's, semi-professional, recreational / leisure-time athletes as well as actual versus former athletes and the age of the before mentioned groups at the time of the manifestation of AF. Please consider that most of recreational marathon runners often participate only once or twice in their life in events like half-marathons or marathons. Please differentiate between different kind of sports and the resulting different (patho-) physiological consequences.

Please consider that typical endurance sports like marathon running, biathlon etc. are not high-intensity sports (line 43 - 45). In opposite to the author's statement such kind of sports are characterized by a low exercise intensity, but long duration or high training volume.

A major critical point is the use of the term "overtraining". In the submitted review there is no clear definition of this pathological status given. The interpretation of an athlete's overtraining as a phenomenon of "extremely high overvoltage" is not scientific based and did not consider the sports medicine / sports science definitions and literature. Therefore please consider that

overtraining is a mean to enhance performance capacity (often used in training camps for up to 3 weeks). overload is a result of the before mentioned overtraining period and is often characterized by a temporary lowered exercise capacity. After a period of recovery / regeneration / tapering (up to 10 days) an enhanced exercise capacity is resulting. the overtraining syndrome is a pathological status of an athlete, which could be divided in the (early) sympathetic form and the (late) parasympathetic form. Both forms are due to different causes and accompanied by different patterns of symptoms.

Considering the before mentioned definitions will have also an impact on the author’s conclusions related to the development of atrial fibrillation, which should be reviewed.

In the chapter “Treatment” beta-blockers are mentioned as a “background” medication. Please consider that beta-blockers are counterproductive in endurance sports due to for example limited heart rate, cardiac output and muscular carbohydrate metabolism during exercise in recreational as well in professional athletes. Also beta-blockers are restricted by the anti-doping rules in several kind of sports.

Author Response

Comments and Suggestions for Authors

The submitted review of Achkasov et al. covers atrial fibrillation (AF) as a frequently type of cardiac complications in actual or former athletes. Unfortunately, the authors did not differentiate different types of athletes and different kind of sports. But both are preconditions for the time and intensity of exercise / training load, which will lead to different kind of cardiac adaptations and (re-) modelling of the cardiac cavities, myocardial walls and tone.

Clear definitions of different types of athletes: Pro's, semi-professional, recreational / leisure-time athletes as well as actual versus former athletes and the age of the before mentioned groups at the time of the manifestation of AF. Please consider that most of recreational marathon runners often participate only once or twice in their life in events like half-marathons or marathons. Please differentiate between different kind of sports and the resulting different (patho-) physiological consequences.

Answer: We agree with the expert reviewer and changed the following: In the article, we tried to emphasize that the effect on the atria and the provocation of AF largely depends on the duration of the loads, this is more than 4500 hours per career and, of course, depends on the age of the patients. Since on the one hand it is associated with the duration of sports, on the other hand it is associated with the appearance of other diseases, including hypertension, impaired glucose tolerance and others. It is very important that the development of AF is not associated with the concept of a professional, but with the duration, intensity of loads and predisposing factors. We emphasized that the risk of developing AF depends on the intensity of the classes. so, in a prospective study when comparing cyclists and golfers. It was demonstrated that AF was not encountered in low-intensity activities. The intensity of physical training, along with their duration, plays a role in the development of predisposition to AF. This conclusion allows us to make a number of studies on the example of the frequency of detecting AF in multidirectional sports. The former cyclists and golfers who continued to train and golfers who developed the qualities of endurance and complex coordination at an average age of 66 years showed significant differences in the registration of AF. In cyclists, AF was detected in 16% of cases, whereas golfers did not have any paroxysms [18]. In the development of AF in athletes, as indicated above, it is not only acquired diseases and genetic factors that influence. The development of AF is affected by the duration of a sports career, the intensity of classes, as well as age. It is important that most authors do not share professional and amateur athletes, since the intensity, nature and duration of the loads are important. On the example of cyclists who continue to train, it was demonstrated that in old age (66 + -7 years) AF occurs 8 times more often than in the control group. This indicates the role of the nature of training (more significant is the effect of training on endurance) and the age of athletes on the frequency of AF development [18]. However, it is obvious that the direct influence of sports and professionalism, including participation in competitions, can have a specific impact on the development of AF. This requires study.

Please consider that typical endurance sports like marathon running, biathlon etc. are not high-intensity sports (line 43 - 45). In opposite to the author's statement such kind of sports are characterized by a low exercise intensity, but long duration or high training volume.

Answer: We agree with the expert reviewer and fixed a bug.

A major critical point is the use of the term "overtraining". In the submitted review there is no clear definition of this pathological status given. The interpretation of an athlete's overtraining as a phenomenon of "extremely high overvoltage" is not scientific based and did not consider the sports medicine / sports science definitions and literature. Therefore, please consider that

overtraining is a mean to enhance performance capacity (often used in training camps for up to 3 weeks). overload is a result of the before mentioned overtraining period and is often characterized by a temporary lowered exercise capacity. After a period of recovery / regeneration / tapering (up to 10 days) an enhanced exercise capacity is resulting. the overtraining syndrome is a pathological status of an athlete, which could be divided in the (early) sympathetic form and the (late) parasympathetic form. Both forms are due to different causes and accompanied by different patterns of symptoms.

Answer: We agree with the expert reviewer. We meant disadaptation syndrome or distress (according to G. Selye). In the article, we replaced overtraining with overtraining syndrome

Considering the before mentioned definitions will have also an impact on the author’s conclusions related to the development of atrial fibrillation, which should be reviewed.

Answer: We agree with the expert reviewer and indeed, for athletes, the most characteristic is not the sympathetic variant, but the parasympathetic variant of AF development, which corresponds to changes in the autonomic regulation in the case of the overtraining syndrome under discussion.

In the chapter “Treatment” beta-blockers are mentioned as a “background” medication. Please consider that beta-blockers are counterproductive in endurance sports due to for example limited heart rate, cardiac output and muscular carbohydrate metabolism during exercise in recreational as well in professional athletes. Also, beta-blockers are restricted by the anti-doping rules in several kind of sports.

Answer: We agree with the expert reviewer. That is why the article states that for athletes, most authors consider surgical treatment rather than drug treatment. Beta blockers, unlike other drugs with an antiarrhythmic effect, have fewer risks of a proarrhythmogenic effect and are less dangerous when used at high loads.

Reviewer 3 Report

Congratulation to the authors for the nice work conducted.

Just to improve the quality of the paper I will recommend some minor changes

Discuss the possible psychological effect of this pathology in the subjects, and how it could limit their activities and normal life

Discuss if new sport modalities as high intensity training and ultramarathon races could be a risk of non-diagnosed athletes

In the treatment point, exist a non-pharmacological intervention for this pathology?

Author Response

Comments and Suggestions for Authors

Congratulation to the authors for the nice work conducted.

Just to improve the quality of the paper I will recommend some minor changes

Discuss the possible psychological effect of this pathology in the subjects, and how it could limit their activities and normal life

Answer: We agree with the expert reviewer. Yes, if it is impossible to restore the sinus rhythm and / or in the presence of clinical manifestations of chronic heart failure, athletes cannot be allowed to engage in sports that require qualities of endurance, strength, speed. According to the European recommendations for admission to competitive activity in 2010, and the position of experts of the American College of Cardiology, dopuc can only be to low-intensity, non-dynamic sports. This significantly changes the lifestyle and motor activity of the athlete and requires not only cardiological, but also psychological support. We added it to the text.

Discuss if new sport modalities as high intensity training and ultramarathon races could be a risk of non-diagnosed athletes

Answer: We agree with the expert reviewer. Most likely, these new sports, as well as the marathon, can be dangerous. The problem under consideration is relevant not only in relation to athletes involved in classic sports, but for professionals and amateurs gaining popularity in high-intensity or requiring great endurance sports. These include ultramarathon, triathlon. This is important, as older athletes are more likely to engage in such sports. Obviously, such athletes must undergo an in-depth examination.

In the treatment point, exist a non-pharmacological intervention for this pathology?

Answer: We agree with the expert reviewer. Yes, surgical treatment is preferred for patients with AF paroxysms. If there is persistent preservation of sinus rhythm, such athletes may cancel anticoagulant therapy.

Round 2

Reviewer 1 Report

I would like to thank the authors for taking into consideration my previous comments and for inserting into the manuscript Table  and Figure 1 in an effort to summarize their points.

However, English of the manuscript is still puzzling whereas rigid organization of data provided is still missing. To give some examples, terminology used for AF in Figure 1 does not coincide with the formal one used (what do authors mean by "private" AF or by prolonged AF<48 hours). Additionally, an example of bad structure is the fact that authors in a subsection about genetic factors predisposing to AF append data about demographic or training parameters.

Finally, even though they have enriched their references they do not seem to fully take advantage of the data included in them. For example I can not see anywhere the importance of AF ablation as a first line treatment for those athletes who will go on with sports participation, while the problems of beta blockade administration in athletes is barely not discussed.

Author Response

Response to reviewer 1

Answer: We agree with the expert reviewer.

I thank you for the great work done to analyze our work. I agree with your comments.

Allow me to answer your comments and questions.

In Figure 1, we had in mind frequent paroxysms and a persistent form of atrial fibrillation or when patients suffer from both paroxysmal and persistent AF episodes.

Long paroxysms of more than 48 hours, these are paroxysms of more than 48 hours and less than 7 days.

Our position takes into account the classification of the European recommendations for the treatment of atrial fibrillation, Reference European Heart Journal (2016) 37, 2893–2962 doi: 10.1093 / eurheartj / ehw210.

We restructured the section “3. Etiology ”and we pointed out that there are many reasons for the development of AF in athletes, some of which relate to known genetic diseases, and some either have no evidence of a genetic etiology or are the result of external or internal effects on the myocardium.

We took into account the comments on the importance of catheter ablation and the treatment of beta AF blockers in athletes and made additions to the chapter on treatment.

Additions in the text are highlighted in green.

Reviewer 2 Report

As already written in the first review one major critical point is the use of the term "overtraining". In the submitted review there is no clear definition of this pathological status given. The interpretation of an athlete's overtraining as a phenomenon of "extremely high overvoltage" is not scientific based and did not consider the sports medicine / sports science definitions and literature. Therefore, please consider that

overtraining is a mean to enhance performance capacity (often used in training camps for up to 3 weeks). overload is a result of the before mentioned overtraining period and is often characterized by a temporary lowered exercise capacity. After a period of recovery / regeneration / tapering (up to 10 days) an enhanced exercise capacity is resulting. the overtraining syndrome is a pathological status of an athlete, which could be divided in the (early) sympathetic form and the (late) parasympathetic form. Both forms are due to different causes and accompanied by different patterns of symptoms.

The authors do not address this in the revised version. The old theory of Seyle only supports general aspects of disadaptation, but not a link between the transient overtraining syndrome and cardiac diseases like AF.

In general, all the other issues, which are mostly well discussed in the author's reply, did not occur in the same manner in the revised version. Therefore please revise it again.

Author Response

Response to reviewer 2

Answer: We agree with the expert reviewer.

Indeed, the violation of the adaptation of the athlete's body to external and internal influences does not have a clear statement. There is no clear concept of pathological sports heart in today's International Classification of Diseases 10 revision. Although a condition is noted there that develops as a result of excessive physical and other influences and it is classified as cardiomyopathy. However, in modern cardiological literature, this term characterizes a sharp decrease in contractility and is not accepted for use in describing the athlete’s heart. Even in international guidelines, atrial fibrillation refers to “Atrial Fibrillation in Athletes,” but a specific condition or diagnosis is not mentioned. This is the subject of study, as well as today's and tomorrow's debate.

We took into account the remark and used the term overtraining syndrome and pathological sports heart.

We took into account the pathophysiological mechanisms of the development of Overtraining syndrome and Nonfunctional syndromes in the arguments and conclusions on the treatment and prevention of pathological sports heart. These considerations do not contradict the article.

As for the features of the development of atrial fibrillation in athletes of various skill levels and various sports, it can be noted that we have not seen multicenter and prospective studies on this topic. However, as indicated in the article, AF is more common among endurance athletes. This differs from the previously existing point of view about the high frequency of pathology in weightlifters. Violation of the rhythm cannot be associated only with the fact that the athlete is a professional or an amateur. AF is associated with a combination of the factors described (training duration, age, intensity, the presence of adaptation disorders, the presence of infections ...). Since non-professionals often train on their own, without control, the risk of rhythm disturbances is much higher for professional athletes.

Edits in the text are highlighted in blue.

Round 3

Reviewer 1 Report

I would like to thank authors for taking into considerations all of my comments and suggestions which I feel have improved the content of the manuscript. However, I still insist that the manuscript strictly needs adequate English language editing. Unfortunately, throughout the manuscript the language and syntax used are rather confusing.